# State-of-the-Art Treatments for Atrial Fibrillation in Patients with Hypertrophic Cardiomyopathy

**DOI:** 10.3390/jcm10143025

**Published:** 2021-07-07

**Authors:** Diego Penela, Antonio Sorgente, Riccardo Cappato

**Affiliations:** 1Arrhythmia and Electrophysiology Research Center Gruppo Multimedica, Via Milanese 300, 20099 Sesto San Giovanni, Italy; diego.penela@multimedica.it; 2Department of Cardiology, Epicura Hospitalier Centre, 7301 Hornu, Belgium; sorgente.antonio@gmail.com

**Keywords:** atrial fibrillation, hypertrophic cardiomyopathy, antiarrhythmic drugs, catheter ablation, anticoagulation

## Abstract

Atrial fibrillation (AF) and hypertrophic cardiomyopathy (HCM) are two very common clinical entities, which often occur simultaneously, giving a hard time to both patients and cardiologists. Myocyte hypertrophy, myocyte disarray and interstitial fibrosis in the left atrium (LA) predisposes to atrial arrhythmias due to modifications of the substrate that promote re-entry. AF is usually poorly tolerated due to the shortening of the diastolic time with rapid heart rates and the lack of the atrial contribution to the diastolic filling in patients who often have a previous diastolic dysfunction. AF onset frequently results in exercise intolerance and recurrent heart failure admissions and also has prognostic implications. Early maintenance of sinus rhythm appears as a worthy approach in these patients, especially when started early in the course of the disease. However, treatment with antiarrhythmic (AA) agents in HCM patients is less effective than in patients without the disease, and concerns regarding safety frequently limit the long-term adherence. Catheter ablation has limited efficacy in patients with persistent AF but can play an important role in patients with paroxysmal AF, emphasizing the importance of an accurate patient selection. The aim of this review is to provide an overview of the pathophysiology of combined HCM and AF and the principal pharmacological and non-pharmacological treatments recommended in this complex clinical scenario.

## 1. Introduction

Atrial fibrillation (AF) is the most frequent arrhythmia in patients with hypertrophic cardiomyopathy (HCM). It is estimated that the prevalence of atrial fibrillation ranges between 20% and 30% in HCM patients. Several reasons can explain this relationship. Myocardial histopathology in patients with HCM is characterized by extensive myocyte hypertrophy, myocyte disarray and interstitial fibrosis [1]. These findings are not limited only to the ventricle but also affect the left atrium (LA) [2]. Diffuse interstitial atrial fibrosis predisposes to atrial arrhythmias that may promote re-entry secondary to heterogeneity of current conduction, shortening of action potentials, depolarization of resting cardiomyocytes and induction of spontaneous phase 4 depolarisations [3]. In addition, patients with HCM present haemodynamic conditions that may predispose to AF. Among them are increase of left atrial pressure secondary to diastolic impairment, left ventricular outflow tract obstruction (LVOTO) and mitral regurgitation which, alone or in combination, contribute to left atrial dilatation and remodelling.

In patients with HCM, the presence of AF is associated with a substantial risk of heart-failure-related mortality [4]. From a clinical perspective, AF is poorly tolerated due to the lack of the atrial contribution to the diastolic filling. In addition, shortening of the diastolic time with rapid heart rates leads to increased LV filling pressures, which can result in exercise intolerance and recurrent heart failure admissions. Finally, the risk of stroke in HCM patients developing AF is substantially higher than in the general population, and risk stratification accuracy of the most commonly used scores remains suboptimal in this population [5].

Treating AF in the setting of HCM is probably one of the most challenging tasks in clinical practice. The aim of this paper is to provide a review of the state of the art on treatment of AF in patients with HCM.

## 2. AF Anticoagulation Strategy

Patients with HCM who experience AF carry a significantly increased risk of stroke. A systematic review [6] described a 27% prevalence of thromboembolism in these patients, with an estimated incidence of thromboembolic events of 3.75% per 100 patients per year. Unfortunately, in these patients, the CHADS_2_ and CHA_2_DS_2_-VASc score is unable to identify patients at low risk, as outlined by a 10% rate of thromboembolic events reported in patients with HCM with CHA_2_DS_2_-VASc 0 during a 10-year observation time [7]. Conversely, other characteristics such as age, left atrial diameter or baseline LVOT obstruction have been suggested to predict thromboembolic events in HCM patients. Nevertheless, evidence is insufficient to identify reliable predictors of AF and thromboembolism. Consistent with these observations, current guidelines recommend lifelong anticoagulation therapy in all patients with HCM and AF regardless of the CHADS_2_ and CHA_2_DS_2_-VASc score [8].

The optimal oral anticoagulation approach for HCM patients is yet to be established. At present, no randomized trials have been conducted to address this question. In addition, the number of patients with HCM included in non–vitamin K antagonist oral anticoagulant (NOAC) trials is insufficient for performing conclusive subgroup analysis to evaluate the comparative benefit of these therapies in this population.

Observational studies have documented no differences in efficacy between NOAC and vitamin K antagonist therapy, although some trend towards lower major bleeding rates in the former therapy group has been shown [9]. Data extracted from a nationwide Korean database identifying 2397 patients with HCM and non-valvular AF receiving chronic oral anticoagulation suggest a reduction in ischaemic stroke in the group on NOACs [10]. Based on the current evidence, NOACs are considered a worthy option in HCM patients with AF. However, randomized trials would be welcome for answering this relevant demand.

## 3. Lifestyle Modification

Patients with AF and HCM should receive similar treatment options and recommendations to those for patients without HCM. This includes the treatment of potential triggers of AF such as obesity, hypertension, hypercholesterolaemia and diabetes and sleep apnoea [11]. However, patients with HCM are commonly limited in their physical health and are more likely to develop obesity, hypertension, hypercholesterolaemia and metabolic syndrome than the general population [12]. Consistent with this finding, more than two-thirds of HCM patients included in a recent retrospective analysis were overweight or obese [13]. The cause of obesity in this population appears to be multifactorial, although restriction in physical activity associated with the diagnosis of HCM plays a significant role. The new internationally released guidelines on the management of HCM [8] recommend mild to moderate intensity recreational exercise in patients with HCM in order to improve cardiovascular performance and fitness and to reduce the occurrence of the classical cardiovascular risk factors. Beyond physical inactivity, some comorbidities predisposing for AF seem to be more frequent in HCM than in the general population. Hypertension prevalence was reported to be as high as 50% in HCM patients, whereas sleep-disordered breathing prevalence ranged from 55% to 70%. In view of this common concomitance, an active ruling out of these morbidities seems to be justified after a new AF diagnosis. Finally, tobacco and alcohol intake can increase the probability of AF occurrence [14]. Alcohol should be consumed with moderation, especially in patients with LVOT obstruction, as it can increase the left atrial pressure by increasing the degree of intraventricular obstruction [15]. Nevertheless, epidemiological data suggest a lower tobacco and alcohol intake in HCM patients than in the general population, which may reflect a psychosocial adjustment to a chronic heart condition [16].

## 4. Pharmacological Rhythm Control

The selection of a rhythm control strategy in patients with HCM is guided by two main considerations. Firstly, loss of sinus rhythm commonly leads to a rapid deterioration of functional class and increased access to hospital admission due to heart failure. Secondly, an early rhythm control strategy may interrupt the vicious circle of loss of atrial function, LA overload and atrial remodelling with potential implications on the ability to restore sinus rhythm in the long term [17].

Pharmacological rhythm control is usually the first option in HCM patients (Figure 1). Treatment with antiarrhythmic drugs (AADs) in these patients is less effective than in patients without HCM. At present, there are no randomized trials investigating the efficacy and safety of the different AADs in patients with AF and HCM.

Figure 1 summarizes the therapeutic management of atrial fibrillation in patients with HCM. In these patients, lifelong anticoagulation therapy is recommended regardless of the CHADS_2_ and CHA_2_DS_2_-VASc score. Ruling out and treating potential triggers of AF such as obesity, hypertension, hypercholesterolaemia, diabetes and sleep apnoea should be considered in all patients. A rhythm control strategy should be the first option, whereas rate control should be considered in patients who have failed several attempts towards rhythm control or those with a low probability of sinus rhythm maintenance (i.e., severe left atrial dilatation). Amiodarone and sotalol are the most commonly used drugs for rhythm control, with the former being the most effective but sotalol being better tolerated (considered as the first option in younger patients). In case of left ventricular outflow tract obstruction, myectomy should be considered, if indicated. Catheter ablation should be considered as an effective option in selected patients, but usually, concomitant administration of antiarrhythmic drugs is required to maintain sinus rhythm.

Amiodarone is the most frequently used drug in this clinical setting. The preference for this drug can be explained by its higher efficacy compared with that of other AADs in other populations [18]. Sotalol also appears as a favoured option in HCM patients. A recent retrospective analysis by Miller and co-workers [19] investigated the efficacy of AADs during 12 years of treatment, where most patients were being treated with amiodarone and sotalol. Both drugs were effective in reducing AF episodes, although sotalol was better tolerated in the long term. These results suggest that sotalol could be a preferred option, especially in younger patients. A recent case series from the Cleveland Clinic [20] reveals that dofetilide is also usually well tolerated in patients with AF and HCM, facilitating management of AF in 84% of the included patients. Even if on-treatment QT interval prolongation was observed, no life-threatening proarrhythmogenic effects were reported in this series. In spite of the promising results, more evidence regarding safety is needed to draw definitive conclusions.

Other AADs are particularly indicated in specific clinical settings. Disopyramide has been shown to reduce the LVOT gradients in two-thirds of patients with LVOT obstruction without a significant increase in the ventricular arrhythmic events [21]. As a result, disopyramide may be indicated in symptomatic, obstructed HCM patients with AF. Finally, class IC AADs are used with care and generally not as a primary option in HCM patients due to the concern of a proarrhythmic effect. However, a considerable proportion of patients with HCM are ICD carriers either for primary or secondary prevention. Initial reports on the use of these drugs in patients with HCM confirm the safety of this approach.

## 5. Catheter Ablation

Interrupting the electrical interactions between excitable tissues within the pulmonary veins (PVs) and the remaining atria by means of catheter ablation remains a cornerstone therapy in patients with AF. Regrettably, use of this strategy shows worse results in HCM patients than in patients with idiopathic AF or with AF in the setting of other diseases. Two observations may explain these results. Firstly, atrial myocyte hypertrophy increases LA wall thickness, thus reducing the likelihood of creating durable transmural lesions. Secondly, HCM is commonly associated with LA dilatation, remodelling and diffuse interstitial fibrosis, thus increasing the risk of elicitation of triggering from extra-PVs foci.

Evidence regarding the efficacy of catheter ablation in HCM patients is growing. Castagno et al. [22] recently published the results of a multicentre trial including a total of 111 patients undergoing 1.6 procedures and followed during a 6-year follow-up. At the last available follow-up, 61% of patients were in sinus rhythm. The authors concluded that most patients require re-do procedures, that efficacy was obtained with the concomitant administration of AADs in most patients and that efficacy was time dependent. Three meta-analyses [23,24,25] on AF and HCM have been published in the last 5 years. Table 1 shows a summary of these three studies. Two of them sought to investigate the success rate of catheter ablation in patients with HCM. The third one aimed to compare the efficacy of catheter ablation between patients with HCM and patients without HCM. In summary, the probability of postablation recurrence of atrial arrhythmias was found to be twofold higher in patients with HCM compared with that in controls, after either single or multiple procedures. AADs were found to be paramount in order to maintain sinus rhythm, as shown by a fivefold higher probability of AAD use after ablation in patients with HCM. Nonetheless, the combination of catheter ablation with AADs was associated with a success rate as high as 75% after one procedure.

Cryothermal ablation is an alternative energy source recently introduced to obtain PV isolation, which is enjoying growing popularity in daily practice [26]. Results of cryoablation for HCM with AF are in line with the results of radiofrequency. A retrospective analysis comparing cryoballoon ablation in patients with and without HCM [27] showed a one-year AF-free rate of 63.0% and 79.0% and a two-year AF-free rate of 55.1% and 77.8%, respectively. A recent multicentre report [28] including 137 HCM patients who underwent AF ablation showed no differences in efficacy between cryoablation and RF ablation. This study also showed that maintenance of sinus rhythm is the exception in persistent AF and that left atrial diameter was the most important predictor of recurrence. These findings suggest that the optimal timing for catheter ablation is early in the course of the disease, having a role in patients with paroxysmal AF.

A relevant aspect of AF ablation in patients with HCM relates to the inherent risk of periprocedural complications. Although such risk appears to be superimposable on those observed in the general population, an incidence between 3% and 4.8% of PV stenosis has been reported. It is difficult to understand whether this finding is related to a higher propensity for HCM patients to develop a hyperactive response to radiofrequency or is just related with the low number of patients with HCM included in these studies. In a recent large-scale, real-world study drawn from the U.S. National Inpatient Sample during the years between 2003 and 2015 [29], one complication occurred in at least 16% of the patients with a mortality rate of 1%. A learning-curve effect was noted, since cardiac and pericardial complications decreased from 8.8% to 2.3% and from 2.8% to 0.9%, respectively, when comparing the first phase of the study with the latest one. Based on these results, the authors advise caution when considering catheter ablation of AF in patients with HCM.

## 6. AF Treatments in Patients with LVOT Obstruction

In case AF occurs in the context of an obstructive phenotype of HCM, treatment for LV obstruction should have the priority, as AF occurrence can act in this case as a marker of haemodynamic impairment caused by the obstructive condition. According to the recently published international guidelines on HCM [8], patients with LV obstruction should receive non-vasodilating beta blockers or non-dihydropyridine calcium channel blockers (e.g., verapamil, diltiazem). In case of persistence of symptoms, a combination with disopyramide should be considered. If medical therapy is not sufficient to control symptoms, referral to specialized tertiary centres for septal reduction therapy should be considered. Removal of the cause of LV obstruction by means of myectomy or septal ablation may lead to a reduction of the burden of AF and its clinical impact on the disease. Furthermore, recent data have shown that surgical ablation through a MAZE procedure added to the surgical myectomy can ensure freedom from AF in 64% of patients at 5-year follow-up [30]. AF occurrence after septal reduction seems to also have prognostic implications, as postoperative AF after myectomy identifies a subgroup of patients with worse cardiovascular outcomes [31].

## 7. Rate Control Strategy

Even if a rhythm control strategy should be preferential in HCM patients with symptomatic AF, a rate control strategy is required in a significant proportion of patients. In the clinical practice, rate control is usually considered in patients who have failed several attempts towards rhythm control or in patients who are unsuitable for AF ablation. Nevertheless, this strategy can be considered as the initial approach in patients with a low probability of sinus rhythm maintenance, such as those with long-standing AF or severe left atrial dilatation. First-line therapies include beta blockers and non-dihydropyridine calcium channel blockers, with the former being specially indicated in the case of LVOT obstruction. The role of digoxin in HCM patients is controversial due to its positive inotropic effect that, from an academic point of view, could exacerbate LVOT obstruction. However, in non-obstructive patients, digoxin is frequently used in daily clinical practice. Finally, an ablate-and-pace strategy should be the last option in refractory cases [8].

## 8. Conclusions

AF can affect 25% of patients with HCM and is associated with relevant clinical implications. Usually, AF onset results in the deterioration of functional class and quality of life and frequently exacerbates heart failure symptoms. In addition, AF is associated with a substantial risk of heart-failure-related mortality. Recent evidence suggests that a rhythm control strategy can be related with better outcomes in this population. Treatment with AADs in these patients is less effective than in patients without HCM. Appropriate selection of the specific AAD can overcome this limitation. Amiodarone and sotalol are the most frequently used AADs in this clinical setting, the former being the most effective and the latter being better tolerated. Dofetilide is also well tolerated and recent data suggest a high efficacy during long-term follow-up. Catheter ablation is less effective in HCM than in the general population. However, when performed in selected patients, this therapy is associated with maintenance of sinus rhythm in up to 2/3 of patients in association with postablation AAD administration. When AF occurs in the context of an obstructive phenotype of HCM, treatment for LV obstruction should have priority. Disopyramide is especially indicated in this setting, and surgical AF ablation could represent a reasonable option in patients with an indication to LVOT reduction.

## Figures and Tables

**Figure 1 jcm-10-03025-f001:**
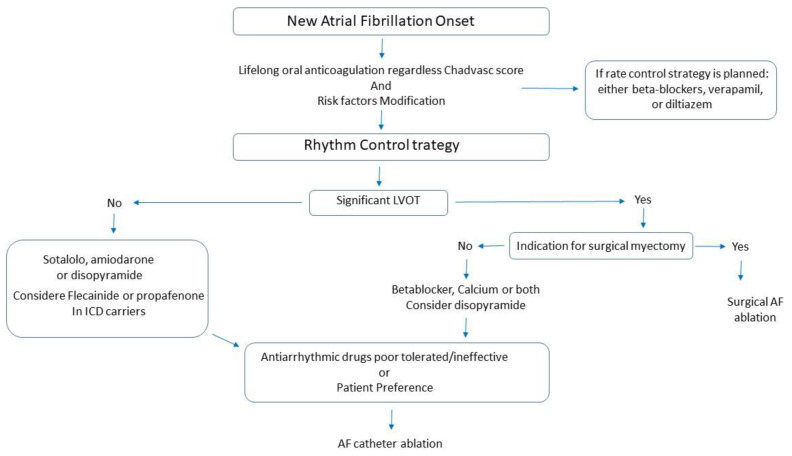
Flowchart. Proposed flow line for handling a new atrial fibrillation onset in HCM patients.

**Table 1 jcm-10-03025-t001:** Characteristics of the 3 meta-analyses published untill now on catheter ablation of AF in patients with HCM.

First Author, Year	Inclusion of Controls *	Number of Patients Included	Median Age or Mean Age or Range (Year-Old)	Duration of AF (Years)	LVEF ** (%)	Left Atrial Size (mm) ***	Length of Mean Follow-Up (Range or Mean, Months)	Single Procedure Efficacy of Catheter Ablation (Range, %)	Multiple Procedure Efficacy of Catheter Ablation (Range, %)
*Zhao DS, 2015*	No	531	48.7–65	3–7.3	56.1–65	45.1–52	11.4–54	29–70	41–92
*Ha HSK, 2015*	No	241	48.7–63	3–8	55–71	46–52	5.8–29	53–75	49–77
*Providencia, 2016*	Yes	532 (139 HCM patients)	57	5.9	N/A	47	21.6	N/A	N/A

* Definition of controls: Patients undergoing catheter ablation of AF but not affected by HCM, ** LVEF: Left ventricular ejection fraction, *** Left atrial size: Left anterior diameter assessed in parasternal longitudinal view with echocardiography.

## Data Availability

Not applicable.

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
