# Peer review of "State-of-the-Art Treatments for Atrial Fibrillation in Patients with Hypertrophic Cardiomyopathy"

_jcm, 2021, doi:10.3390/jcm10143025_

Round 1

Reviewer 1 Report

The clarifications of the authors appear adequate. My suggestion is to accept the paper

Author Response

We thank you very much for your comment.

Reviewer 2 Report

In this brief report, Dr. Cappato and colleagues reviewed the state-of-the-art managements for AF in the presence of HCM. Overall, this is a very nice and insightful manuscript. I only have some suggestions for the authors (and perhaps some minor typo corrections):

  • Because the context of this brief report is the treatment for AF, I would suggest the author to change the heading of section 2 perhaps into "AF anticoagulation strategy" or something similar, and section 6 into "AF treatments in patients with LVOT obstruction". 
  • Two other important risk factors of AF are smoking and alcohol consumption. Is there any data regarding the prevalence of AF with HCM in those specific populations (smokers and drinkers)? Is there any difference as compared to AF without HCM? Please discuss in section 3.
  • In addition to obesity and sedentary lifestyle, the authors also mentioned hypertension, hypercholestrolemia, diabetes and sleep apnea but there is no explanation on what is different from "regular" AF without HCM? Please extend the discussion including those risk factors.
  • Please add a section about rate control, especially discussing the role of beta-blockers and digitalis in AF with HCM. 
  • I think it is worth to mention somewhere around lines 107-109 that the study in reference 17 did not observe any major proarrhythmic side effect of dofetilide (e.g, TdP), although QTc prolongation was observed in some patients.
  • I think it is also relevant to discuss the incidence and prognosis of post-operative AF post HCM myomectomy, for example as shown by Tang et al. (PMID: 31588075) and other studies.
  • Please also adapt Figure 1 to accommodate some new information suggested above. 
  • Please include the figure in line with the text (as per MDPI requirement)
  • Please add a figure title and caption containing a few sentences explaining the figure.
  • Line 12: please replace "2" with "two"
  • Line 42: What is FA? please clarify.
  • I am not sure if I understand the second part of this sentence "Finally, the risk of stroke in HCM patients developing AF is substantially higher than in the general population and represents the most commonly used score for risk stratification in patients with HCMPlease clarify and rephrase.
  • Line 49: should be "state-of-the-art"

Round 2

Reviewer 2 Report

Thank you for addressing my previous comments and suggestions. I do have some more minor suggestions to improve the clarity and readability of the manuscript:

  • Regarding this sentence "Finally, tobacco and alcohol intake can increase the probability of AF occurrence" in line 83, I think it would be relevant to cite this paper about alcohol and AF (PMID: 32710981).
  • Please change or remove "luckily" in line 85 as it does not sound scientific. 
  • Lines 105-108: please make it clear that "series" is a "case series". 
  • Line 299: please change "till" with "until"
  • Line 324: Please provide a figure title 
  • Line 324: "in these patients, lifelong ..." add the comma
  • There is something missing in this sentence "A rhythm control strategy should be considered as the first option, unless very low probability of sinus rhythm maintenance (i.e severe left atrial dilatation). " It does not seem to be grammatically correct. Please rephrase. 
  • Line 328: should be "sotalol"
  • Please be consistent on the English style: British or American? The authors used "summarizes" which is American English but also "apnoea" which is British. Please recheck the whole text together with potential typos and grammar mistakes.
  • In general, there are some serious grammarical errors that need to be corrected because they impaired the readability of this manuscript, particularly in the revised parts of the manuscript (highlighted part). Please consult with native English writer

Author Response

This manuscript is a resubmission of an earlier submission. The following is a list of the peer review reports and author responses from that submission.

Round 1

Reviewer 1 Report

This is a nice and concise review on the determinants, clinical consequences and treatment of AF in patients with HCM. The authors have done a good job in providing a synthesis of recent literature. 

My main concern is that the efficacy of catheter ablation is clearly over emphasized by the authors, who quote success rates of 92% after multiple procedures and 75% after the first procedure. The papers they refer to provide a broad spectrum of results, with an average of approximately 50% and 65%, respectively. In a recent report, Castagno et al. presented a 3 center experience with AF ablation in HCM population. A total of 111 patients 
underwent 1.6 procedures and were followed for median of 6 years. At last follow-up, only 61% patients were in SR. It is clear that a) most patients require re-dos; b) most need to maintain AADs, and c) efficacy is time-dependent, with increasing rates of relapses in the first 5-6 years after the procedure. Most failures occur in patients with severe LA dilatation, which unfortunately is common in HCM. I think it is important to provide the right perspective to patients and physician, in order to avoid unrealistic expectations. 

In the intro: it is unclear how myocardial hypertrophy is per se a cause of reduced cardiac output. 

The authors refer repeatedly to "septal" obstruction... which does not exist. The term is outflow obstruction. 

At end of page 1, disopyramide should be mentioned among drugs used for control of LV outflow obstruction. 

Same paragraph: symptoms in obstructive patients are associated with the severity of the gradient, not with the degree of hypertrophy. 

Beginning of page 2: I agree that surgical treatment of obstruction is the priority in obstructive patients with AF. However, MAZE should be combined with the myectomy, to prevent relapses of AF. This is not mentioned; a recente paper by Bolland colleagues (AJC 2020) nicely shows this. 

In my experience, sotalol is disappointing in controlling AF in HCM. Do the authors have direct experience to share? they mention flecainide and propafenone.. any experience with these agents?

Reviewer 2 Report

A short fairly informative  review on an important topic complementing the sketchy guidelines available fro the treatment of AF in HCM. My comments are:

Authors have left unclear whether class I AAD are contra indicated in this condition (flecainide and propafenone are contraindicated because of AF to AFL transformation)  

Despite the limitations of meta-analysis and systematic review comparison between normal and HCM could be better detailed (for example, ref 9 reports on efficacy of RFA for more than one ablation as well as freedom from AF in HCM and control groups and according to their Forrest plot there is no statistical difference in freedom from AT/AF or number of catheter ablation procedures between normal and HCM patients etc)   

Editing of the text is recommended 

Reviewer 3 Report

I read this manuscript with interest as it tackled an important clinical topic. However, the manuscript suffers from serious flaws, which limit its value:

  1. It does not provide significantly more robust information in relation to the recently published AHA guidelines
  2. Some parts are discussed more deeply (ablation), while others such as AAD treatment and surgical ablation are mentioned only briefly. 
  3. It does not discuss the anticoagulation in AF at all
  4. The part on lifestyle modification is very scarce
  5. There is no data on drugs other than AAD 

Some minor issues:

  • English language
  • The use of informal language
  • Lack of reference for some claims throughout the manuscript.

Round 2

Reviewer 3 Report

I still do not think the manuscript brings sufficiently more new data to previously published guidelines or reviews to merit publication. There is no figure or scheme, which could provide a nice, illustrative take home massage.
